# Prolonged Endurance Exercise Adaptations Counteract Doxorubicin Chemotherapy-Induced Myotoxicity in Mice

Insu Kwon [1], Gwang-Woong Go [2], Youngil Lee [3,*,†] and Jong-Hee Kim [1,4,*,†]

1 Research Institute of Sports Science and Industry, Hanyang University, Seongdong-gu, Seoul 04763, Korea; insu0115@hanyang.ac.kr
2 Department of Food and Nutrition, Hanyang University, Seongdong-gu, Seoul 04763, Korea; gwgo1015@hanyang.ac.kr
3 Department of Movement Sciences and Health, Usha Kundu, MD College of Health, University of West Florida, Pensacola, FL 32514, USA
4 Human-Tech Convergence Program, Department of Physical Education, Hanyang University, Seongdong-gu, Seoul 04763, Korea
* Correspondence: ylee1@uwf.edu (Y.L.); carachel07@hanyang.ac.kr (J.-H.K.); Tel.: +1-850-474-2596 (Y.L.); +82-2-2220-1325 (J.-H.K.); Fax: +1-850-474-2106 (Y.L.); +82-2-2220-1337 (J.-H.K.)
† Thease authors contributed equally to this work.

**Featured Application: This study is firstly provided pathological features with a distinct application of the modified Gomori trichrome (MGT) staining for DOX-induced myotoxicity.**

**Abstract:** Doxorubicin (DOX) is a potent chemotherapeutic agent widely used for various types of cancer; however, its accumulation causes myotoxicity and muscle atrophy. Endurance exercise (EXE) has emerged as a vaccine against DOX-induced myotoxicity. However, potential molecular mechanisms of EXE-mediated myocyte protection for the unfavorable muscle phenotype remain unelucidated. In addition, most studies have identified the short-term effects of DOX and EXE interventions, but studies on the prolonged EXE effects used as adjuvant therapy for chronic DOX treatment are lacking. Twelve-week-old adult male C57BL/6J mice were assigned to four groups: sedentary treated with saline (SED-SAL, $n = 10$), endurance exercise treated saline (EXE-SAL, $n = 10$), sedentary treated with doxorubicin (SED-DOX, $n = 10$), and endurance exercise treated with doxorubicin (EXE-DOX, $n = 10$). Mice were intraperitoneally injected with DOX (5 mg/kg) or saline five times biweekly for eight weeks, while a treadmill running exercise was performed. Body composition was assessed and then soleus muscle tissues were excised for histological and biochemical assays. Our data showed that DOX aggravated body composition, absolute soleus muscle mass, and distinct pathological features; also, TOP2B upregulation was linked to DOX-induced myotoxicity. We also demonstrated that EXE-DOX promoted mitochondrial biogenesis (e.g., citrate synthase). However, no alterations in satellite cell activation and myogenesis factors in response to DOX and EXE interventions were observed. Instead, SED-DOX promoted catabolic signaling cascades (AKT-FOXO3α-MuRF-1 axis), whereas EXE-DOX reversed its catabolic phenomenon. Moreover, EXE-DOX stimulated basal autophagy. We showed that the EXE-mediated catabolic paradigm shift is likely to rescue impaired muscle integrity. Thus, our study suggests that EXE can be recommended as an adjuvant therapy to ameliorate DOX-induced myotoxicity.

**Keywords:** doxorubicin; chemotherapy; myotoxicity; endurance exercise; skeletal muscle; proteolytic system; autophagy; body composition; topoisomerase II β; AKT-FOXO3α-MuRF-1

## 1. Introduction

Cancer is a significant public health concern worldwide because its cases continue to rise. Currently, cancer remains an incurable disease, but an opportune chemotherapeutic strategy suppresses cancer growth and thus extends a lifespan. Doxorubicin (DOX) is the

most widely administered chemotherapeutic agent; however, several side effects, such as severe nausea, vomiting, and alopecia are inevitable. Moreover, DOX's toxicity to normal cells has been a primary clinical concern. Specifically, DOX-induced muscle loss and mortality have remained a critical topic among chemotherapy-mediated side effects [1–5].

While pharmacological and nutritional interventions to mitigate DOX-induced myotoxicity have shown inconsistent results, preclinical studies have shown that endurance exercise (EXE) confers protection against myotoxicity in skeletal muscle. For instance, administration of acute endurance exercise (EXE) before an acute DOX treatment protects against DOX-induced myotoxicity [6,7]. However, given that most clinical chemotherapy paradigms consisting of multiple cycles of DOX treatment are administered, it is imperative to investigate whether prolonged EXE as an adjuvant intervention during chronic DOX treatment can appease myotoxicity to simulate a clinical setting of DOX treatment. Moreover, potential protective molecular mechanisms mediated by chronic EXE against DOX-induced myotoxicity remain unknown.

The exact mechanism of DOX-induced myotoxicity in skeletal muscle is not fully understood yet. However, the fact that DOX's random interposition to deoxyribonucleic acid (DNA) in parallel with topoisomerase II$\beta$ (TOP2B) results in damaging DNA in cardiomyocytes sheds light on how DOX causes myotoxicity [8,9]. More specifically, a recent study has shown that DOX-induced downregulation of TOP2B protein expression promotes cell death via increasing oxidative stress in cardiomyocytes [10]; in contrast, another study has reported that deletion of TOP2B prevents the heart against DOX-induced cardiotoxicity [11]. These conflicting reports suggest that more studies are necessary to verify the underlying role of TOP2B in myotoxicity. More importantly, whether DOX-induced TOP2B modulation is linked to myotoxicity in skeletal muscle remains unknown. Besides, no literature is available if EXE-mediated modulation of TOP2B is associated with myocyte protection against DOX-induced myotoxicity.

Satellite cells are skeletal muscle-specific stem cells crucially contributing to repairing muscle damage and injury. Activation of myogenic regulatory factors (MRFs), such as paired box 7 (PAX7), myogenic differentiation 1 (MYOD 1), and myogenin (MYOG) are necessary for satellite cells to be proliferated and differentiated into mature myotubes [12,13]. Recent evidence has shown that DOX-induced dysregulation of MRFs is associated with myocyte regenerative dysfunction, but acute EXE-induced MRF restoration improves regenerative capacity against DOX [14], suggesting that EXE-induced satellite cell amelioration may be a potential protective mechanism against DOX. However, whether chronic EXE elicits the same protective mechanisms against DOX-induced myotoxicity remains unknown.

Accelerated protein degradation due to DOX administration has been suggested as a potent element in DOX-induced muscle atrophy. In this regard, the ubiquitin-proteasome system (UPS) plays a crucial role in proteolysis. The UPS consists of E1 ubiquitin-activating enzyme, E2 ubiquitin conjugation, and E3 ubiquitin ligase, among which the E3 ubiquitin ligase facilitates ubiquitination to a target protein for degradation by the 26S proteosomes. Therefore, modulations of E3 ligase expression have been implicated in various models of muscle atrophy. Specifically, two muscle-specific E3 ligases, such as muscle atrophy F-box (MAFbx)/atrogin-1 and muscle ring-finger-1 (MuRF-1) in the skeletal muscle have been identified as potent inducers of muscle atrophy [15]. For example, MuRF-1 overexpression increases ubiquitination of myofibrillar proteins and induces muscle atrophy [15]. Similarly, DOX-mediated MuRF-1 upregulation promoted muscle atrophy; in contrast, when MuRF-1 was suppressed by an in vitro electrical stimulation in the cells treated with DOX, cellular atrophy was mitigated [16]. This previous study has provided an interesting question of whether EXE-mediated myocyte protection against DOX-induced muscle atrophy in an *in-vivo* model is associated with MuRF-1 modulation.

In addition to the UPS, autophagy, a lysosome-dependent catabolic process, is involved in protein degradation. Autophagy is critical to sustaining normal cell function by removing damaged proteins, lipids, and small organelles, such as mitochondria, but dysregulated autophagy can contribute to cell death. Recent studies have shown that DOX-induced

autophagy dysregulation leads to muscular dysfunction and oxidative stress, whereas DOX-induced autophagy restriction reverses those impairments [17]. Similarly, a recent study has shown that acute DOX administration increases autophagy and contributes to oxidative stress and reinforced proteolysis; this study also shows that acute exercise preconditioning (DOX administration immediately after five days of 60 min treadmill running/day) prevents DOX-induced autophagy and results in skeletal muscle protection [7], suggesting that EXE-induced autophagy inhibition may be a potential protective mechanism against DOX. However, it should be noted that cellular outcomes from autophagy responses to acute exercise and chronic exercise may differ because some studies have shown that chronic EXE-induced autophagy is necessary for myocyte protection, especially in the heart [18,19]. Thus, it is important to investigate how chronic EXE provided as an adjuvant therapy during the multiple chemotherapy cycles modulates autophagy in skeletal muscle.

Thus, the present study hypothesized that the adjuvant chronic EXE intervention during the multiple DOX treatment would be a potent therapeutic strategy to attenuate DOX-induced myotoxicity.

## 2. Materials and Methods

### 2.1. Animals

Mature adult 12-week-old male C57BL/6J mice (DBL company, Chungcheongbuk-do, South Korea) were used in this study. All mice were housed in a temperature (22 °C) and humidity (55%)-controlled facility at the Center for Laboratory Animal Sciences, under semi-specific pathogen-free (SPF) conditions with a 12:12 h light-dark cycle. The mice were given standard chow (PicoLab® Rodent Diet 20, #5053, LabDiet) and distilled water *ad libitum*. Figure 1A depicts an overview of the experimental design; briefly, the animals were randomly divided into four groups after a one-week adaptation period: sedentary group treated with saline (SED-SAL, *n* = 10), endurance exercise group treated with saline (EXE-SAL, *n* = 10), sedentary mice treated with doxorubicin (SED-DOX, *n* = 10), and endurance exercise group treated with DOX (EXE-DOX, *n* = 10). The body weights of all mice were measured weekly until the end of the experiments.

### 2.2. Doxorubicin Treatment

Animals assigned to DOX treatment groups were intraperitoneally injected with DOX hydrochloride (#D1515, Sigma-Aldrich, St. Louis, MO, USA) dissolved in 0.9% saline at a 5 mg/kg dose, five times biweekly (total 25 mg/kg); in contrast, non-DOX treatment groups were injected with an equivalent volume of 0.9% saline as a vehicle.

### 2.3. Endurance Exercise Training

Mice assigned to the exercise training groups initiated running on a motorized mouse treadmill at a speed of 8–12 m/min for the adaptation period (30 min/day consecutively for five days) and continuously performed the treadmill running at a speed of 12–15 m/min for the main exercise (60 min/day, for seven weeks). The exercise protocol was based on an earlier in vivo study demonstrating its potency to improve muscle repair and maintenance against DOX chemotherapy-induced myotoxicity [20]. Specifically, mice assigned to the non-exercise training groups were placed in the same training room while mice assigned to the EXE groups were running on the treadmill to exclude possible confounding factors (i.e., treadmill noise and lagging in the lane of the treadmill). Moreover, several bristle brushes were used at the end of each lane instead of electrical shock to minimize suffering in mice.

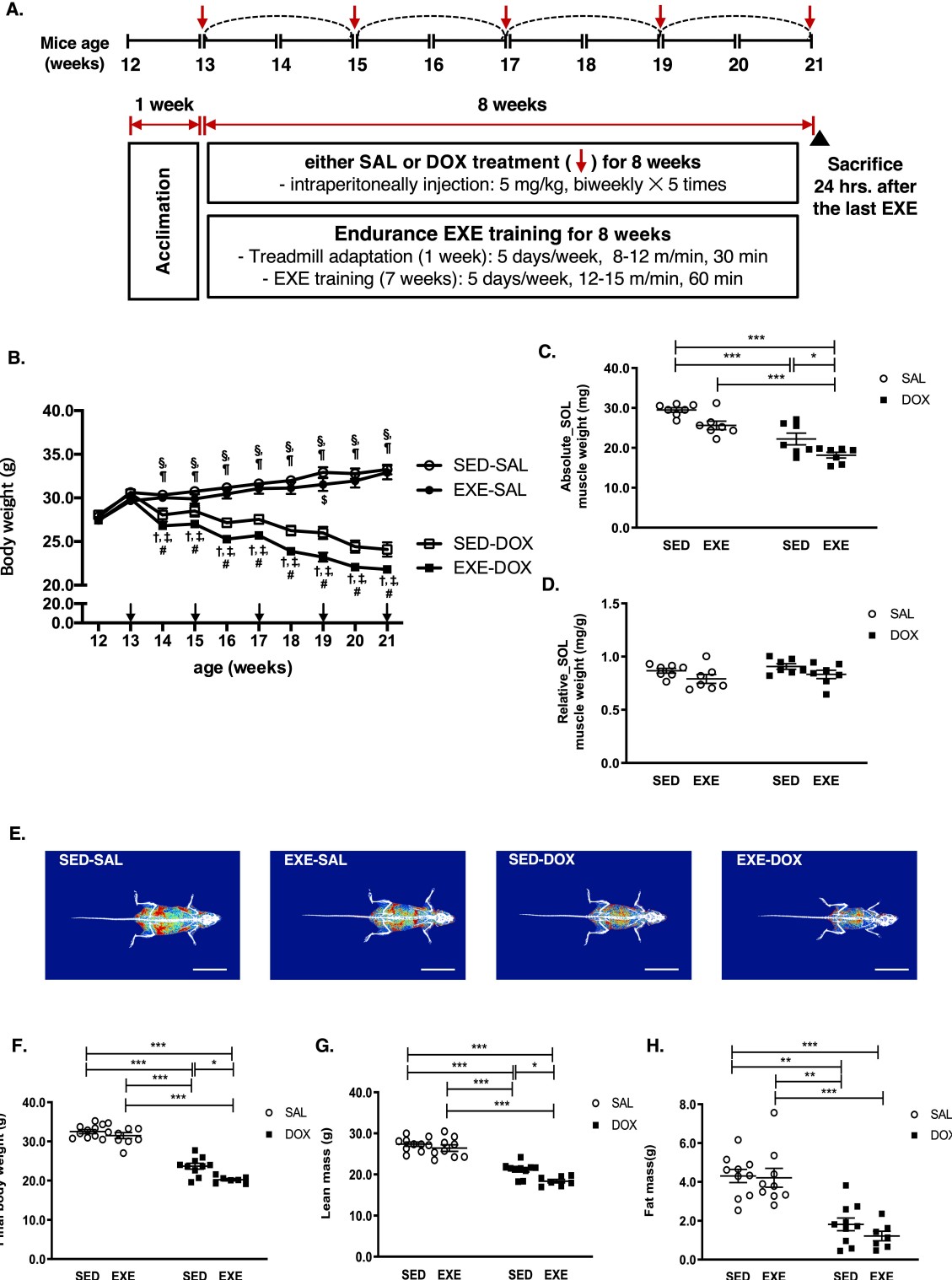

**Figure 1.** DOX-induced body composition and muscle mass changes. (**A**) Weekly body weight changes (weeks × group): DOX treatment gradually reduces body weight beginning from one week after DOX injection. The EXE-DOX shows a more significant body weight reduction than the SED-DOX. The following symbols denote significant differences: SED-SAL vs. EXE-SAL ($), SED-DOX (†), and EXE-DOX (‡); EXE-SAL vs. SED-DOX (§) and EXE-DOX (¶); SED-DOX vs. EXE-DOX (#).

(**B**) Absolute SOL muscle weight for each group. SED-DOX and EXE-DOX exhibit a significant decrement in absolute SOL muscle mass. It is noted that the loss of SOL muscle mass is more significant in the EXE-DOX than in SED-DOX. (**C**) Relative SOL muscle weight. (**D**) Representative DEXA-scanned images of lean tissue (blue) and fat tissue (red). (**E–H**) Quantitative data of final body weight, lean mass, and fat mass, respectively, were acquired by the DEXA scan. Significant differences are denoted by an asterisk: $p < 0.05$ (*), $p < 0.01$ (**), and $p < 0.001$ (***). Values are presented as mean $\pm$ SEM. SED: sedentary; SAL: saline; DOX: doxorubicin; EXE: endurance exercise; SOL: soleus; DEXA: Dual-energy X-ray absorptiometry.

### 2.4. Dual Energy X-ray Absorptiometry (DEXA)

Body composition was evaluated via dual-energy X-ray absorptiometry (DEXA) to compare the distribution of lean and fat tissues during DOX chemotherapy and EXE training. The DEXA analysis was performed 24 h after the drug and exercise interventions for eight weeks, using a body composition analyzer (InAlyzer, MEDIKORS, Gyeonggi-do, South Korea). Before scanning, the mice were anesthetized with pentobarbital sodium (40 mg/kg, intraperitoneal injection) and then laid down on the positioning paper in the DEXA instrument. After finishing their body scan with a high resolution, designated parameters (i.e., final body weight, lean mass, and fat mass) were assessed.

### 2.5. Tissue Collection

The soleus (SOL) muscles were selected as representative muscles in this study since slow-twitch muscle is more affected by DOX treatment than fast-twitch muscle in a dose-dependent manner [21,22]. The left side of the SOL muscle was immediately frozen in liquid nitrogen and stored at $-80\ °C$ for Western blot assay. The right side of the SOL muscle was mounted with optimal cutting temperature (OCT) freezing medium, rapidly frozen in isopentane pre-cooled with liquid nitrogen, and used for histological analysis.

### 2.6. Hematoxylin and Eosin (H&E) Staining

Freshly reserved SOL muscle tissues were sectioned at 3 μm in thickness using a sliding cryotome (Leica CM-1860) and mounted on microscope slides. Then, the tissue sections were briefly incubated in hematoxylin (#H3136, Sigma-Aldrich) and washed in tap water sequentially for 5 min. The slides were consecutively incubated in 1% hydrochloric acid (#H1758, Sigma-Aldrich) and 70% ethanol for 1 min and then washed in tap water for 5 min. After that, the slides were incubated in an eosin Y (#E4009, Sigma-Aldrich) for 2 min. The stained tissue slides were successively immersed in 70%, 80%, and 90% ethanol for 1 min each, after which the sections were incubated in 100% xylene (#534056, Sigma-Aldrich) and 100% ethanol (#51976, Sigma-Aldrich) mixture (1:1 ratio) for 1 min and then dipped into 100% xylene for 3 min. Mounting was performed using the VECTASHIELD Mounting Medium (#H-1400, Vector Laboratories, Newark, CA, USA). Finally, the representative images were captured at 200× magnification using a slide scanner (Axio Scan.Z1, ZEISS, Jena, Germany). Image J software was used to measure the cross-sectional area (CSA) and numbers of myofibers per pixel.

### 2.7. Modified Gomori Trichrome (MGT) Staining

Cross-sectioned (3 μm) SOL muscle tissues were fully dried at ambient temperature and incubated in Mayer's hematoxylin solution for 10 min and washed with water for 5 min. Then, the tissue sections were stained with filtered Gomori trichrome solution for 20 min and washed with 0.2% glacial acetic acid solution for 3 min three times, respectively. The tissues were successively dehydrated using ethanol in a series of concentrations (80%, 90%, and 100%) and cleared with 100% xylene. Finally, the tissue sections were mounted using the VECTASHIELD Mounting Medium (#H-1400, Vector Laboratories), and images were captured by a slide scanner (Axio Scan.Z1, ZEISS) and examined.

### 2.8. Protein Extraction and Western Blot Analysis

The SOL muscles were homogenized (1:10 *w/v*) in T-PER® Tissue Protein Extraction Reagent (#78510, ThermoFisher Scientific, Waltham, MA, USA) containing a Halt™ protease and phosphatase inhibitor cocktail (100×) (#78440, ThermoFisher Scientific, Waltham, MA, USA) with a homogenizer stirrer (#HS-30E, DAIHAN Scientific, Seoul, Korea) using a tissue grinder (#SL.cw.011.102, SciLab® Korea, Seoul, Korea). Following the manual instructions, the homogenates were centrifuged at $10,000 \times g$ for 15 min at 4 °C in a Sorvall Legend Micro 17R Centrifuge (#75002444, ThermoFisher Scientific, Waltham, MA, USA) to obtain the upper cytosolic fraction. Protein concentration was assessed using the Pierce™ Coomassie protein assay kit (#23200, ThermoFisher Scientific, Waltham, MA, USA). The supernatants were heated in a mixture of $4 \times$ Bolt™ LDS sample buffer (#B0008, Invitrogen, Waltham, MA, USA) and $10 \times$ Bolt™ sample reducing agent (#B0009, Invitrogen, Waltham, MA, USA) at 70 °C for 10 min and cooled on ice for 10 min. Equal amounts of 40 μg of the protein were loaded onto Bolt™ 4–12% Bis-Tris Plus Gels, 15-well (#NW04125BOX, Invitrogen) for target proteins, except for LC3B, which was separated with Bolt™ 12% Bis-Tris Plus Gels, and onto 15-well (#NW00125BOX, Invitrogen) for 1 h at room temperature (RT) in diluted $20 \times$ Bolt™ MOPS Running Buffer (#B000102, Invitrogen, Waltham, MA, USA). After electrophoresis, the gels were transferred in diluted $20 \times$ Bolt™ Transfer Buffer (#BT00061, Invitrogen, Waltham, MA, USA) to 0.45 μm nitrocellulose membranes (#1620115, Bio-Rad) or 0.2 μm PVDF membranes (#1620112, Bio-Rad, Hercules, CA, USA) and used to a hydrophobic and low molecular weight protein (LC3B) for 1 h at RT. The membranes were briefly stained with Ponceau S solution (#P7170, Sigma-Aldrich, St. Louis, MO, USA) to confirm the successful transfer and rinsed with pure water until the background is disappeared. Non-specific proteins were blocked for 60 min at RT in 5% non-fat milk or bovine serum albumin (BSA) with Tris-buffered saline containing 0.1% tween-20 (TBS-T). The membranes were then incubated with the designated primary antibodies overnight at 4 °C. The primary antibodies used were as follows: topoisomerase II-beta (TOP2B, #25330), citrate synthase (CS, #390693), paired box 7 (PAX7, #81975), myogenic differentiation 1 (MYOD1, #377460), myogenin (MYOG, #12732), muscle atrophy F-box (MAFbx/atrogin-1, #166806), and muscle ring finger protein-1 (MuRF-1, #398608) from Santa Cruz Biotechnology; AKT (#9272), p-AKT$^{Thr308}$ (#13038), p-AKT$^{Ser473}$ (#4060), mammalian target of rapamycin (mTOR, #2983), p-mTOR$^{Ser2448}$ (#5536), forehead box O3α (FOXO3α, #12829), p-FOXO3α$^{Ser253}$ (#13129), Beclin-1 (BECN-1, #3495), ATG7 (#8558), and p62 (#23214) from Cell Signaling Technology; Cathepsin L (CTSL, #133641) from Abcam; LC3B (#L7543) from Sigma-Aldrich. The next day, the membranes were washed twice for 5 min in $1 \times$ TBS-T and then incubated with the designated secondary antibodies conjugated with HRP for 1 h at RT. The secondary antibodies were goat anti-rabbit IgG (#G-21234) and goat anti-mouse IgG (#62-6520) from ThermoFisher Scientific. Next, the membranes were washed thrice for 5 min in $1 \times$ TBS-T, and immunoreactive proteins were detected using Amersham ECL western blotting detection reagents (#RPN2209, Cytiva). The band images were acquired using a ChemiDoc™ XRS⁺ Imaging System (#1708265, Bio-Rad), and the band intensities were quantified using the Image Lab™ software (version 6.0; #1709690, Bio-Rad). We used a Ponceau S staining method as a suitable alternative loading control to conventional housekeeping proteins, such as actin, GAPDH, and tubulin because total protein staining methods (e.g., Ponceau S and stain-free gel) have been justified to be superior or better than beta-actin in previous studies [23,24]; we also noticed that GAPDH was subject to change in response to long-term EXE. Total proteins on the membrane were stained with Ponceau S. The intensity of the proteins between 100 and 37 kDa in each lane was quantified via densitometry Image Lab™ software (version 6.0; #1709690, Bio-Rad). Then the intensity of protein expression of interest was normalized by the intensity of the Ponceau S-stained total proteins in each corresponding lane. Ponceau S–stained images showing protein bands between 75 and 37 kDa were cropped and provided evidence of loading controls in each Western blot figure because of space limits. Data were presented as percent of fold-changes.

## 2.9. Statistics

All data analyses were conducted using the GraphPad Prism® software (version 6.0; San Diego, CA, USA). Data are presented as mean ± SEM. To compare the body weight changes between and within groups, two-way repeated measures of analysis of variance (ANOVA) were used. When a significant difference was notified, a Tukey's honestly significantly different (HSD) test was used as a *post hoc* to find group differences. A two-way ANOVA was used for all other dependent variables; when appropriate, a Tukey's HSD *post hoc* was used to determine group differences. Statistical significance was set at $p < 0.05$.

## 3. Results

### 3.1. DOX Causes Loss of Body Weight, Lean Mass, Fat Mass, and Absolute SOL Muscle Mass

The onset of body weight loss due to DOX treatment (SED-DOX and EXE-DOX) was observed one week after the first DOX injection compared to saline injection (SED-SAL and EXE-SAL). The significant loss of body weight in the SED-DOX and DOX-EXE groups compared to the SED-SAL and EXE-SAL groups continued for the following seven weeks (weeks 15 through 21) (Figure 1B). Moreover, the EXE-DOX group lost significantly higher body weight than the SED-DOX group during the treatment periods.

The absolute SOL muscle weight was lower in the SED-DOX and EXE-DOX groups than in the SED-SAL group, and the loss of muscle mass was more significant in the EXE-DOX group than in the SED-DOX group (Figure 1C). However, the relative SOL muscle weights (SOL muscle weight to body weight) were not different among the groups (Figure 1D).

To investigate whether the loss of body weight was due to the reduction of lean mass, fat mass, or both, we assessed the body composition of mice using DEXA. The representative whole body scan images of each group displayed lean tissue (blue) and fat tissue (red) (Figure 1E). The DEXA results showed that mice's final body weight, lean mass, and fat mass were reduced in the SED-DOX and EXE-DOX groups compared to the SED-SAL and EXE-SAL groups (Figure 1F–H); it was notable that the extent of loss of the body weight and lean mass was more significant in the EXE-DOX group compared to the SED-DOX group.

### 3.2. EXE Mitigates DOX-Induced Myotoxicity

We examined the histological phenotypic characteristics of SOL muscle. H&E staining data showed that the SED-DOX group accumulated degenerating fibers, whereas the EXE-DOX group mitigated myocyte degeneration (Figure 2A). Ragged red fibers, a hallmark of muscle pathology due to the accumulation of dysfunctional mitochondria, can be visualized with the MGT staining as a red rim in the subsarcolemmal area. Our data showed that DOX treatments (both SED-DOX and EXE-DOX) increased ragged red fibers; however, the EXE-DOX group exhibited fewer ragged red fibers compared to the SED-DOX group (Figure 2B,C). The CSA of myofibers in the SOL muscle was significantly reduced in all three groups compared to the SED-SAL group (Figure 2D); however, the CSA did not differ between the SED-DOX and EXE-DOX groups. The number of myofibers was increased in all three groups compared to the SED-SAL group (Figure 2E) due possibly to the reduction of the CSA.

### 3.3. DOX Causes DNA Damage but Does Not Disrupt Satellite Cell Activation and Myogenesis

Since DOX-induced TOP2B upregulation is associated with cardiotoxicity [11], we examined for the first time if EXE would downregulate TOP2B and thus protect myotoxicity. TOP2B levels were higher in the SED-DOX group than in the SED-SAL and EXE-SAL groups; however, their levels were not statistically different among the SED-SAL, EXE-SAL, and EXE-DOX groups (Figure 3A,B). The CS levels have been used as a mitochondrial biogenesis indicator. Our data showed that CS levels were significantly increased in the EXE-DOX group compared to the SED-SAL and SED-DOX groups (Figure 3A,C); interestingly, EXE per se (EXE-SAL) did not promote CS upregulation.

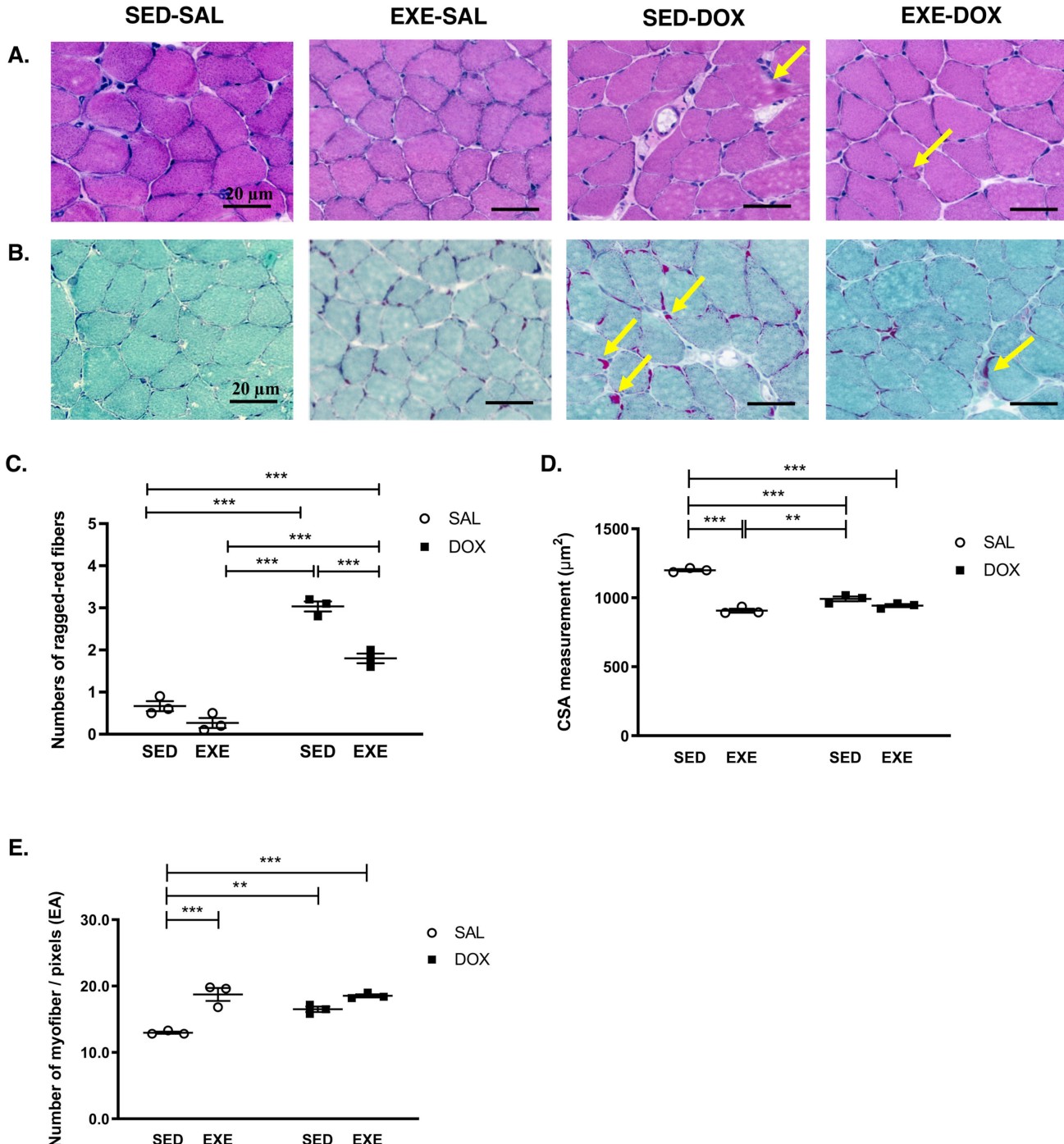

**Figure 2.** EXE-induced myocyte protection against DOX-induced myotoxicity in soleus muscle. (**A**) Hematoxylin and eosin (H&E) staining. The arrows show degenerating fibers. EXE-DOX shows protection against myocyte degeneration (*n* = 3/group). (**B**) Modified Gomori Trichrome (MGT) staining (*n* = 3/group). The arrows indicate pathological diagnostic of ragged red fibers. Total magnification is 200×, and the size of a scale bar is 20 μm. (**C**) Quantification of ragged-red fibers/designated area. (**D**) Measurement of the cross-sectional area (CSA) of myofibers. (**E**) Numbers of myofiber per designated area. Asterisks denote significant differences: *p* < 0.01 (**), and *p* < 0.001 (***). Values are presented as mean ± SEM. SED: Sedentary; SAL: saline; DOX: doxorubicin; EXE: endurance exercise.

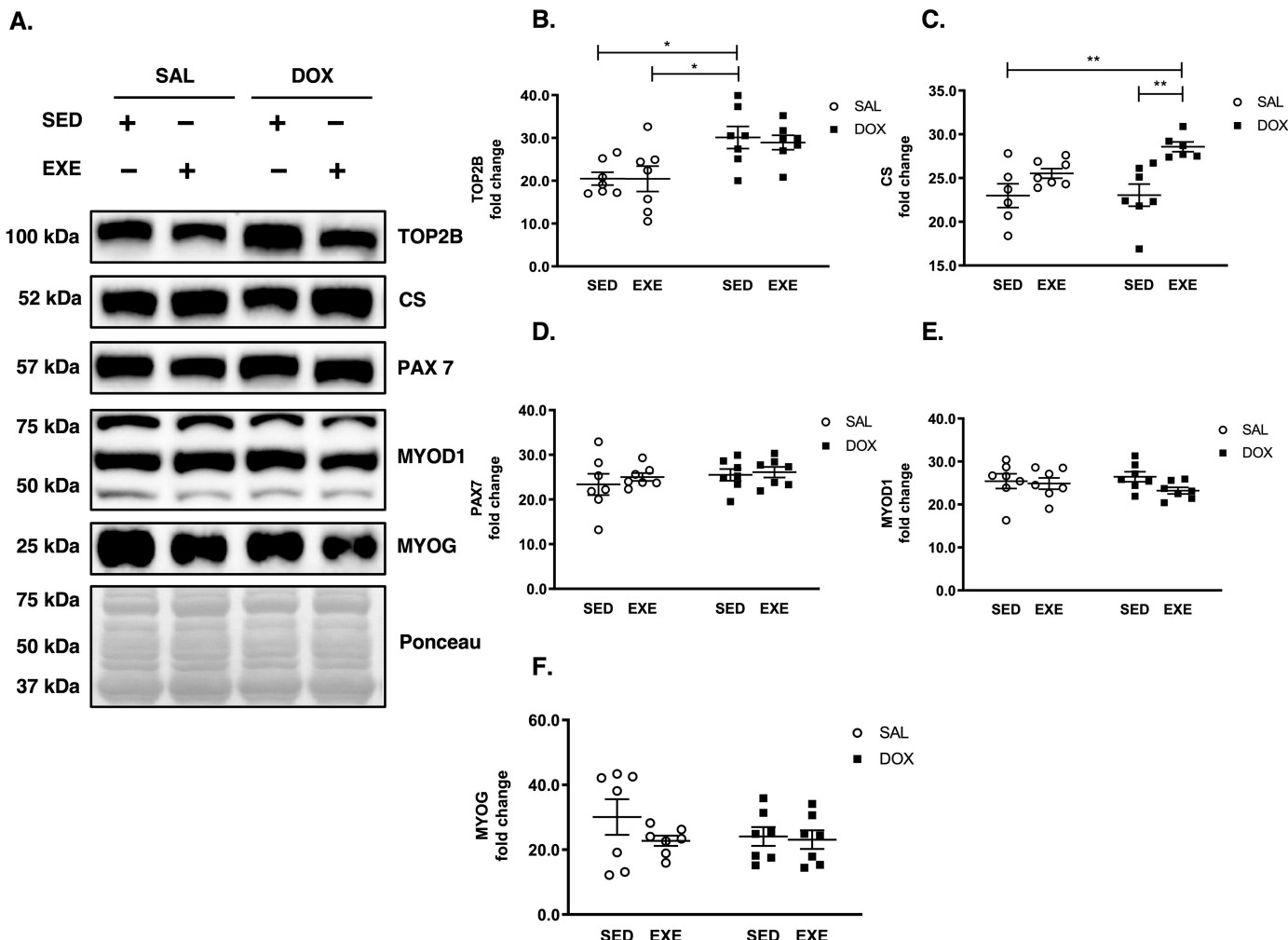

**Figure 3.** Effects of EXE on DNA damage, satellite cells activation, and myogenesis-related markers in soleus muscle against DOX treatment. (**A**) Representative western blot images. (**B**) Quantification of TOP2B protein expression. Both SED-DOX and EXE-DOX significantly upregulate TOP2B compared to the SED-SAL. (**C**) Quantification of CS protein expression. EXE-DOX significantly increases CS compared to the SED-SAL and the SED-DOX. (**D–F**) Quantification of PAX7, MYOD1, and MYOG protein expression, respectively. All target proteins were normalized by Ponceau-stained total proteins ($n$ = 6–7/group). Asterisks denote significant differences: $p < 0.05$ (*) and $p < 0.01$ (**). Values are presented as mean ± SEM. SED: Sedentary; SAL: Saline; DOX: Doxorubicin; EXE: Endurance exercise; TOP2B: Topoisomerase II-beta; CS: Citrate synthase; PAX7: Paired box 7; MYOD1: Myogenic differentiation 1; MYOG: Myogenin.

Next, we measured the markers of satellite cells activation (PAX7) and myogenesis (MYOD1 and MYOG). Neither SED-DOX nor EXE (EXE-SAL and EXE-DOX) treatments modulated the expressions of those markers (Figure 3A,D–F).

### 3.4. EXE Prohibits DOX-Induced Proteolytic Activation

To examine if DOX-induced loss of SOL muscle mass is associated with dysregulations between protein synthesis and breakdown, we explored the pivotal axis of the protein synthesis signaling pathway (AKT and mTOR). The p-AKT$^{Ser473}$/AKT ratio was significantly upregulated in the EXE-DOX group compared to other groups (Figure 4A,B). Interestingly, no changes in the p-mTOR$^{Ser2448}$/mTOR ratio were observed among groups (Figure 4A,C). Next, we examined myostatin (MSTN), also known as growth differentiation factor-8 (GDF-8), and its downstream target SMAD 2/3. MSTN levels were reduced in the EXE group compared to other groups (Figure 4A,D), and SMAD 2/3 levels were not changed among groups (Figure 4A,E). We further investigated FOXO3α-mediated proteolytic signaling pathways, including MAFbx/atrogin-1 and MuRF-1. Concomitant with the AKT results above, the ratio of p-FOXO3α $^{Ser253}$/FOXO3α was higher in the EXE-DOX group than in the SED-DOX group. The ratio also was increased in the EXE-SAL group compared to the SED-SAL group and the SED-DOX group (Figure 4F,G). We next assessed FOXO3α target genes MAFbx/atrogin-1 and MuRF-1. While MAFbx/atrogin-1 protein levels were not different among groups (Figure 4F,H), MuRF-1 levels were significantly elevated in the SED-DOX group compared to the other three groups (Figure 4F,I).

### 3.5. EXE Promotes Basal Autophagy

Autophagy is another cellular catabolic process and examined by measuring autophagy induction protein BECN-1, autophagosome formation proteins ATG7 and LC3B-II, and autophagy flux protein p62 and lysosomal protease CTSL. Our results showed that BECN-1 and ATG7 expression levels did not differ among groups (Figure 5A–C). On the contrary, LC3B-II and an LC3B-II/I ratio were elevated in the EXE-DOX group compared to other groups (Figure 5A,D), but neither p62 nor CTSL differed among groups (Figure 5A,E,F).

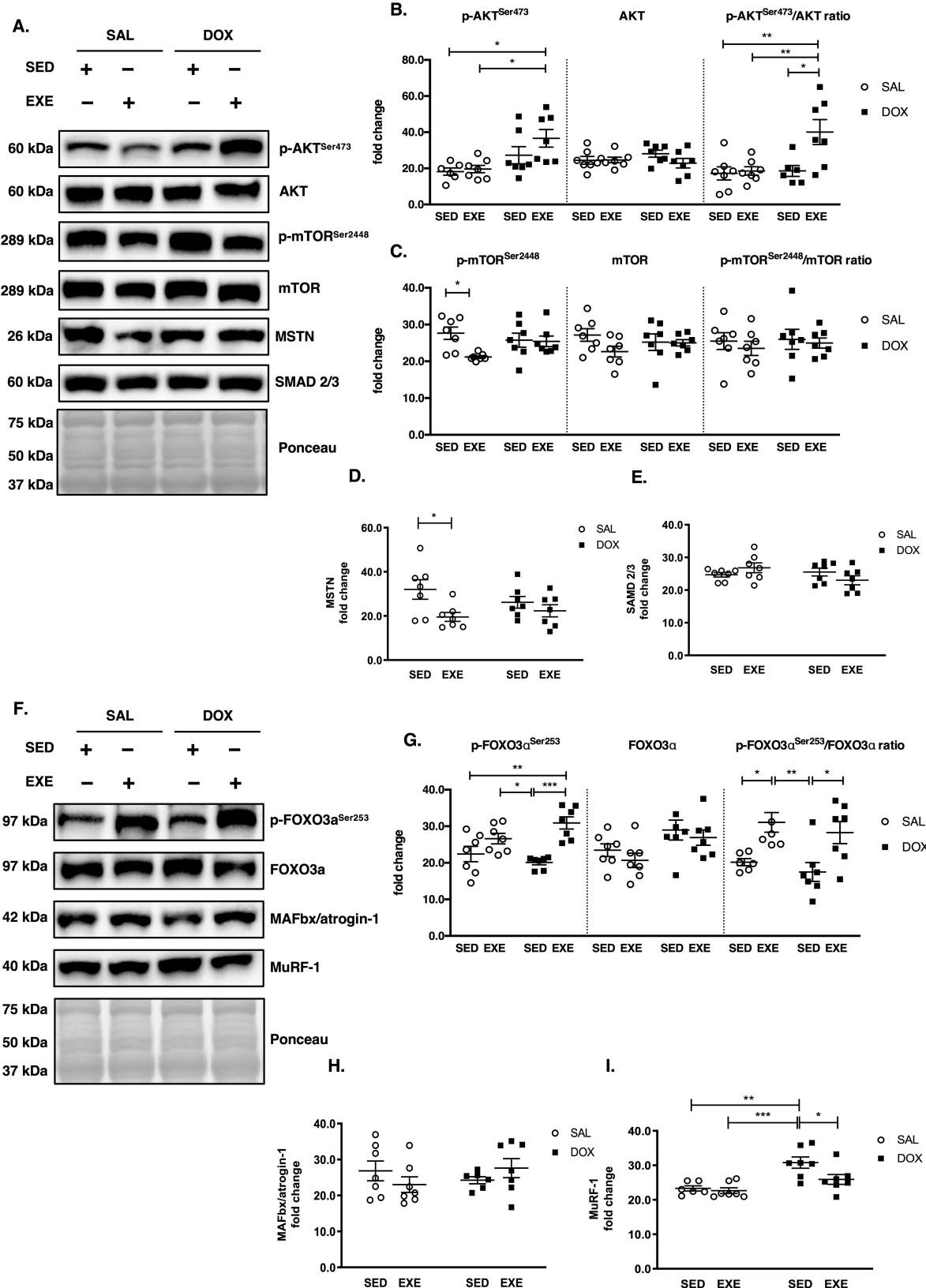

**Figure 4.** Effects of EXE on DOX-mediated proteolysis signaling pathways in soleus muscle. (**A**) Representative western blot images of anabolic and proteolytic signaling proteins. (**B**) Quantification of total protein expression and phosphorylation levels of p-AKT$^{Ser473}$, AKT, and p-AKT$^{Ser473}$/AKT ratio. (**C**) Quantification of total protein expression and phosphorylation levels of p-mTOR$^{Ser2448}$,

mTOR, and p-mTOR$^{Ser2448}$/mTOR ratio. (**D,E**) Quantification of MSTN and SMAD 2/3 protein expression, respectively. (**F**) Representative western blot images of proteolytic proteins. (**G**) Quantification of total protein expression and phosphorylation levels of p-FOXO3$\alpha^{Ser253}$, FOXO3$\alpha$, and p-FOXO3$\alpha^{Ser253}$/FOXO3$\alpha$ ratio. (**H,I**) Quantification of MAFbx/atrogin-1 and MuRF-1 protein expression, respectively. All target proteins were normalized by Ponceau-stained total proteins ($n = 6$–$7$/group). Asterisks denote significant differences: $p < 0.05$ (*), $p < 0.01$ (**), and $p < 0.001$ (***). Values are presented as mean $\pm$ SEM. SED: Sedentary; SAL: Saline; DOX: Doxorubicin; EXE: Endurance exercise; mTOR: Mammalian target of rapamycin; Forehead box O3$\alpha$: FOXO3$\alpha$; MAFbx: Muscle atrophy F-box; MuRF-1: Muscle ring finger protein-1.

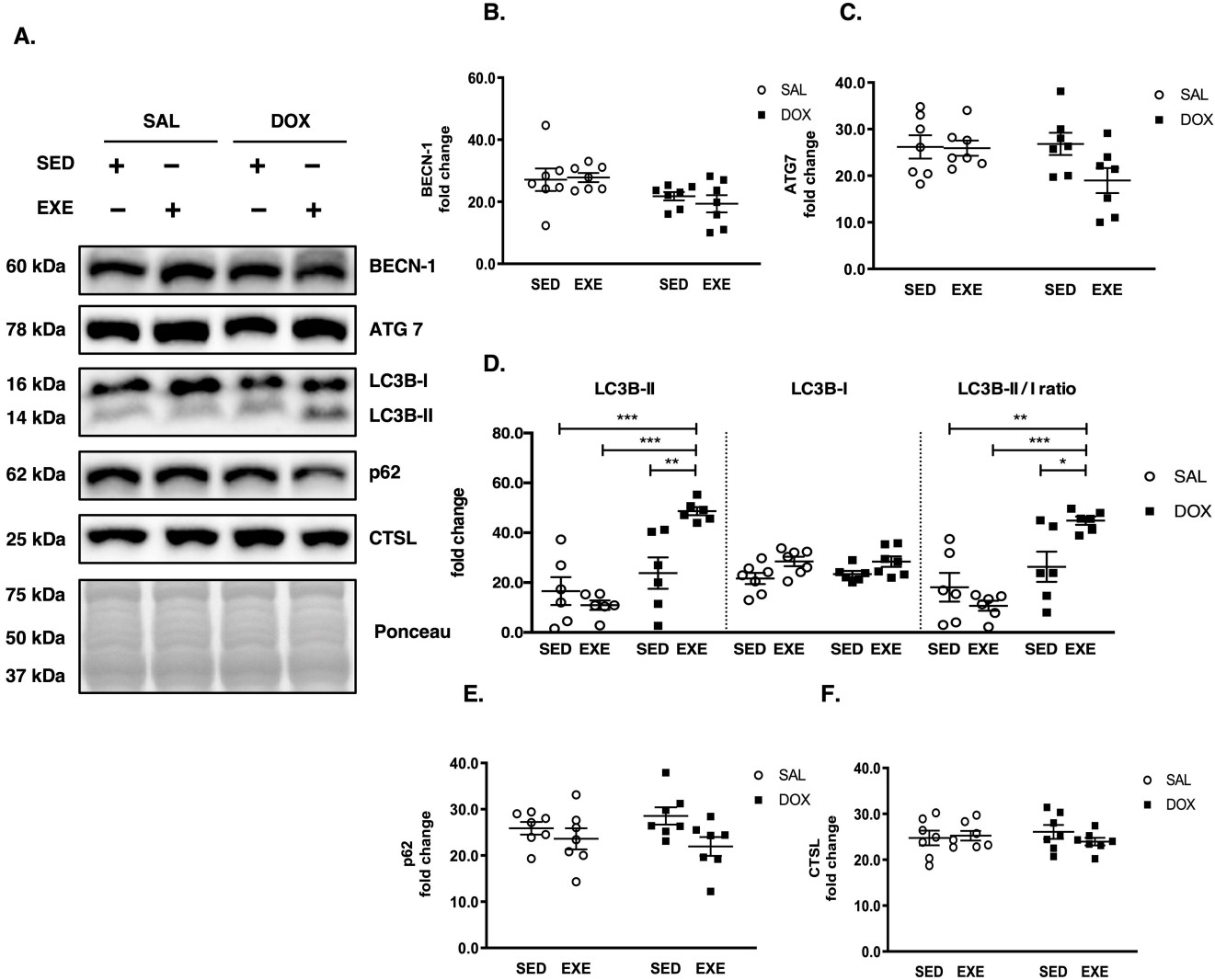

**Figure 5.** Effects of EXE on autophagy against DOX treatment in soleus muscle. (**A**) Representative western blot images of autophagy proteins. (**B**) Quantification of BECN-1 protein levels. (**C**) Quantification of ATG7 protein levels. (**D**) Quantification of LC3B-II and LC3B-I protein levels and LC3B II/I ratio. (**E**) Quantification of p62 protein levels. (**F**) Quantification of CTSL protein levels. All target proteins were normalized by Ponceau-stained total proteins ($n = 6$–$7$/group). Asterisks denote significant differences: $p < 0.05$ (*), $p < 0.01$ (**), and $p < 0.001$ (***). Values are presented as mean $\pm$ SEM. SED: Sedentary; SAL: Saline; DOX: Doxorubicin; EXE: Endurance exercise BECN-1: Beclin-1; CTSL: Cathepsin L.

## 4. Discussion

Despite being a highly effective anticancer agent, DOX administration is limited by its potent toxicity in various tissues, including skeletal muscle. Recent studies have shown that acute EXE before DOX treatment protects DOX-induced myotoxicity; however, it remains unclear whether EXE should be administered as an adjuvant intervention during multiple DOX treatment cycles to appease unwanted toxicity to non-cancer tissues. Thus, the present study examined the effect of chronic EXE administration as an adjuvant treatment during chronic DOX treatment cycles on phenotypic changes in SOL muscle. The present study provides novel insight into the mechanisms responsible for EXE-induced myocyte protection against chronic DOX treatment cycles. Our data showed that EXE administered during DOX treatment did not prevent DOX-induced loss of body weight, lean mass, and absolute SOL muscle mass, but sustained healthier muscle morphology. Furthermore, our results showed that suppression of proteolytic activation induced by the AKT-FOXO3$\alpha$-MuRF-1 axis and potentiation of basal autophagy is linked to the EXE-mediated muscle quality control against DOX-induced myotoxicity.

Previous studies have shown that DOX treatment leads to an unfavorable phenotype in skeletal muscle tissues, such as muscle atrophy and myotoxicity [22,25–27]. Our data also verified those anomalies as evidenced by loss of body weight, absolute SOL muscle weight, and body composition (reduction of final body weight, lean mass, and fat mass examined by DEXA) in response to DOX treatment (5 mg/kg, biweekly, total dosage = 25 mg/kg). Consistent with our findings, a recent study also showed a significant reduction in body weight and gastrocnemius mass in response to chronic DOX treatment (2.5 mg/kg, twice a week, total dosage = 25 mg/kg) [28]. Interestingly, EXE intervention combined with DOX treatments resulted in a more significant loss of body weight and absolute SOL muscle mass than the only DOX treatment as opposed to our postulation that EXE might be attenuated DOX-induced muscle wasting. We are not sure why this incidence happened in our study; however, considering the possibility that DOX incurs a loss of appetite and that EXE increases caloric expenditure, the combinatory effects of DOX and EXE may elicit the additional loss of body weight and muscle mass.

The next question is, does this observation mean that cancer patients receiving DOX chemotherapy should refrain from EXE during the DOX chemotherapy period due to additional muscle loss? According to our results, the answer is "no" because the relative SOL muscle mass was indistinguishable between SED-DOX and EXE-DOX groups, and a muscle damage level (e.g., the appearance of degenerated muscle fibers and ragged red fibers) was lower in the EXE-DOX group compared to the SED-DOX group. Consistent with our affirmation, other studies have confirmed that various EXE paradigms provide similar therapeutic potency against DOX-induced myotoxicity. For instance, short-term EXE (<4 weeks) before DOX treatment [7,14,29–31], long-term EXE ($\geq$4 weeks) before DOX treatment [4], short-term EXE (<4 weeks) during/after DOX treatment [28,32,33], and long-term EXE ($\geq$4 weeks) during/after DOX treatment [5,20,34] conferred myocyte protection. Furthermore, a recent clinical study has shown that cancer patients performing EXE during chemotherapy improve a disease-free survival rate compared to sedentary cancer patients [35], indicating that EXE administration during multiple chemotherapy cycles is safe and beneficial.

A DOX-mediated TOP2B modulation plays a crucial role in myotoxicity, as the ablation of TOP2B attenuates DOX-induced DNA damage in MEF cells [36]. Besides, cardiac-specific deletion of TOP2B protects the heart against DOX-induced cardiac toxicity [11], demonstrating that DOX-induced TOP2B upregulation is linked to DOX-induced myotoxicity. However, it remains unknown whether TOP2B is involved in DOX-induced myotoxicity in skeletal muscle, particularly in oxidative SOL muscle. In this regard, our results for the first time showed that DOX-induced TOP2B upregulation was associated with myotoxicity. Based on this observation, we postulated if EXE-mediated TOP2B hindrance might be associated with myocyte protection against DOX. However, our postulation was rejected since EXE treatment failed to prevent DOX-induced TOP2B upregulation. Thus, it seems



reasonable for us to exclude the TOP2B scenario from possible protective mechanisms of EXE against DOX in SOL muscle at least.

Recent studies have demonstrated that DOX-induced mitochondrial damage provokes cellular dysfunction [17]. Given the notion that EXE enhances mitochondrial function and promotes mitochondrial biogenesis [37,38], we postulated that EXE-mediated mitochondrial enrichment might rescue myocytes from DOX toxicity. To confirm the alterations of mitochondrial volume, we measured the CS levels since they have been used as a prime index of mitochondrial volume estimation, especially in oxidative muscle [39,40]. We found that SED-DOX treatment did not affect the mitochondrial volume; more surprisingly, EXE-SAL (8 weeks of treadmill running for 60 min/day at 70–80% VO$_2$max) failed to increase the CS levels (mitochondrial volume), which contradicts the canonical concept of EXE as a potent inducer of mitochondrial biogenesis. Nevertheless, EXE-DOX significantly increased the CS levels, suggesting that EXE may promote mitochondrial biogenesis upon DOX toxicity to accelerate mitochondrial turnover (more biosynthesis of mitochondria in parallel with selective removal of injured mitochondria) for myocyte protection. Indeed, this presumption corresponds to our histological data shown in Figure 2A,B, showing EXE-induced myocyte protection.

Besides mitochondria, MRFs have been reported to be involved in DOX-induced myotoxicity. A previous in vitro study has shown that downregulation of MRFs in C2 myoblasts in response to DOX administration impairs myogenic differentiation [41], suggesting that restoration of DOX-induced MRFs loss would be a potential protective mechanism. In support of this concept, a recent study using rats has implicated that EXE-induced upregulation of MRFs may protect skeletal muscle against DOX treatment [14]. However, our study does not support the above study's finding because EXE did not alter MRFs, such as PAX7 (activation of satellite cells), MYOD1 (commitment to differentiation), and MYOG (fusion into myotube). This conflicting result may be due to different study designs in which we used a chronic EXE protocol (eight weeks of treadmill exercise whereas the other study used an acute EXE protocol (two weeks of treadmill exercise). Moreover, the mode of DOX treatment could have affected the results: a chronic model of DOX treatment (e.g., five injections on a bi-weekly basis during which EXE was administered) vs. an acute model of DOX (e.g., one injection immediately after two weeks of EXE). Although these seemingly different study designs might cause the discrepancy, further studies are warranted to define whether EXE-induced MRFs modulation is necessary for myocyte protection against DOX-mediated myotoxicity.

Disequilibrium between protein synthesis and proteolysis is a primary determinant of muscle atrophy. In this regard, growing evidence has shown that DOX-induced proteolysis contributes to muscle atrophy due to the unrestrained ubiquitin-proteasome system and calpain activities [29,42] in which E3 ligase MuRF-1 expedites myofibril degradation [16]. Moreover, disrupted protein synthesis caused by DOX has been reported to contribute to muscle atrophy [28,43]. The present study postulated that EXE-induced reversion of DOX-induced dysregulation of protein synthesis and degradation would be associated with myocyte protection. Increased FOXO3a activation (lower levels of phosphorylated FOXO3a) would indicate an increase in proteolysis because dephosphorylated FOXO3a can translocate to the nucleus to induce MuRF-1 expression. As displayed in Figure 4F,G, phosphorylated FOXO3a levels were lower in the SED-DOX group than in the EXE-DOX group. This means that the SED-DOX maintained active FOXO3a. As a result, MuRF-1 levels were also increased in the SED-DOX group, suggesting more proteolysis. Contrarily, FOXO3a phosphorylation (inactivated FOXO3a by AKT) levels were higher in the EXE-DOX than in the SED-DOX. As a result, MuRF-1 levels were significantly lower in the EXE-DOX than SED-DOX, suggesting less muscle loss in the EXE-DOX group. Although the result seems to be insightful that EXE is involved in mitigating DOX-induced proteolysis, this EXE-induced anti-proteolytic response contradicts our observation that EXE exacerbated DOX-induced SOL muscle loss (Figure 1C,G). Thus, we postulated that EXE might disrupt

protein synthesis, leading to more muscle loss. This proposition was not the case because neither SED-DOX nor EXE-DOX altered a p-mTOR$^{Ser2448}$ state.

We next were prompted to seek to explain why EXE-DOX promoted more muscle loss but maintained healthy muscle morphology from autophagy because autophagy is another catabolic process comparable to the UPS. While the UPS degrades short-lived and soluble proteins, autophagy catabolizes long-lived and insoluble proteins via lysosome-dependent pathways to maintain skeletal muscle homeostasis [44–47]. Recent studies have shown that dysregulated autophagy is associated with skeletal muscle atrophy and myopathy [48–50]. In this context, it has been reported that an acute DOX administration (a single dose at 20 mg/kg) upregulates autophagy and concurs with muscle injuries in the rat's SOL muscle [17,29]. On the contrary, the inhibition of DOX-induced autophagy, using an adeno-associated virus overexpressing dominant-negative ATG5, or EXE obstructs DOX-induced soleus muscle atrophy and oxidative damages [17,29]. These studies suggest that acute DOX-induced autophagy may contribute to muscle atrophy and that EXE-induced anti-autophagy is necessary to protect myocytes.

Interestingly, however, as opposed to an acute DOX treatment, a chronic DOX treatment did not provoke autophagy [5], and EXE-induced autophagy instead conferred cardiac protection against a chronic DOX treatment [19]. Consistent with these studies, we also observed that chronic DOX treatment did not induce autophagy and EXE promoted autophagy. From this present study, we cannot expound on the distinct autophagy responses to acute vs. chronic DOX treatment and acute EXE vs. chronic EXE against DOX; however, autophagy may be a tentative adaptive response to acute DOX treatment rather than chronic pathologic manifestation. As such, chronic EXE-induced autophagy promotion may contribute to myocyte protection since it facilitates a selective removal of long-lived damaged proteins and small organelles, such as dysfunctional mitochondria.

In this regard, our findings showing that healthier muscle morphology and upregulation of basal autophagy (e.g., EXE-induced LC3B-II upregulation without p62 accumulation) against chronic DOX treatments suggest that EXE-induced basal autophagy may be a critical element involved in myocyte protection. Furthermore, we presume that the EXE-induced basal autophagy may be responsible for the loss of SOL muscle mass rather than MuRF1-dependent mechanisms observed in DOX-treated SOL muscle. To confirm whether EXE-induced autophagy is essential against chronic DOX treatment, the use of a genetically modified mouse model of autophagy, specifically inhibiting EXE-induced autophagy (e.g., a BCL2 AAA mouse model), [51] is warranted in the future study.

## 5. Conclusions

Our study reports that chronic DOX treatment weakened body composition indices (final body weight, lean mass, and fat mass) and an absolute SOL muscle mass. Interestingly, the adjuvant EXE treatment exacerbated the body composition indices; nevertheless, EXE mitigated DOX-induced myotoxicity. This chronic EXE-induced protective response was associated with suppression of FOXO3α-mediated proteolytic signaling pathways (e.g., inhibition of FOXO3α and reduction of its target gene MuRF-1) and improved basal autophagy. Our findings in a mouse model provide encouraging evidence that EXE can be recommended as an adjuvant therapy to appease myotoxicity and enhance the quality of muscle against repetitive cycles of DOX chemotherapy.

### Limitations

The potential limitation in the present study is that although it would be best practice to use tumor-bearing mice to simulate the physiological conditions of cancer patients, the present study chose healthy mice instead to examine the molecular mechanisms of DOX-induced myotoxicity in normal healthy skeletal muscle. Thus, it is possible that the results from nontumor-bearing mice would not represent the actual cancer physiology occurring in cancer patients.

**Author Contributions:** I.K., G.-W.G., Y.L. and J.-H.K. participated in the design of the study; I.K. and J.-H.K. conducted animal handling, biochemical assays, and data collection; I.K., G.-W.G., Y.L. and J.-H.K. contributed to data analysis and interpretation of results. All authors contributed to the manuscript writing. All authors have read and approved the final version of the manuscript and agreed with the order of presentation of the authors. All authors have read and agreed to the published version of the manuscript.

**Funding:** This work was supported by the Ministry of Education of the Republic of Korea and the National Research Foundation of Korea (2018S1A5B5A02037951) and the Cooperative Research Program for Agriculture Science and Technology Development (a supportive managing project of the Center for Companion Animals Research, #PJ0147592022) and the Rural Development Administration of the Republic of Korea.

**Institutional Review Board Statement:** All procedures were approved by the Institute of Animal Care and Use Committee (IACUC) of Hanyang University (approval no. 2019-0039A). The mice were maintained in accordance with the guidelines.

**Informed Consent Statement:** Not applicable.

**Data Availability Statement:** Not applicable.

**Acknowledgments:** We would like to thank the faculty members of Hanyang Medical Research Supporting Center for providing animal care and technical assistance.

**Conflicts of Interest:** The authors declare no conflict of interest.

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
