# Peer review of "Prolonged Endurance Exercise Adaptations Counteract Doxorubicin Chemotherapy-Induced Myotoxicity in Mice"

_applsci, doi:10.3390/app12073652_

Round 1

Reviewer 1 Report

In the manuscript, titled “Prolonged endurance exercise adaptations counteract doxorubicin chemotherapy-induced myotoxicity in mice”, the authors investigate the effects of exercise on mice repeatedly dosed with doxorubicin, specifically as it relates to muscle phenotype and muscle mass. While interesting, the presented data seem to contradict each other, making interpretation of the results of this study difficult. As written, this study does not reveal mechanism, and instead only suggests at some myo-protective effects that are not fully supported by the authors’ protein expression data. Perhaps some of these nuances are missing simply from a lack of context of the results, but as written the study is difficult to interpret and the impact is missing, which significantly reduced this reviewer’s enthusiasm.

Major Concerns

  1. Section 2.1 references a Figure that is not included in the manuscript draft.
  2. In Section 3.1, “it was notable that the extent of loss of the body weight and lean mass was more significant in the EXE-DOX group compared to the SED-DOX group” is unclear. Are the authors stating that there is a significant decrease in mass in EXE-DOS vs. SED-DOX (based on the reading of the figure)?
  3. Are the animals the same age/size in Figure 1D? There are significant differences in the length/size (including skeletal structure) of the animals, which does not seem to make sense.
  4. Are the CSA measurements of the entire soleus muscle, which is implied in the text (Section 3.2)? If so, including the average CSA of the myofibers would also be of interest.
  5. Quantification of the number of “ragged” fibers (Fig 2B) is necessary to determine the protective effects of exercise. While the relative increase in CS expression may indicate mitochondrial biogenesis, quantification and/or evidence of protective effects are missing. Could the authors assess the relative number of mitochondria in each muscle?
  6. Much of the presented data seem to be contradicting each other. The authors suggest that increased FOXO3a activation would indicate an increase in proteolysis (which supports loss of muscle mass and autophagy), however then MuRF-1 expression is also diminished (which supports less muscle loss). The authors also present evidence of autophagy in the presence of DOX-EXE, but little evidence of that is visualized histologically. The authors need to supply more discussion and/or evidence to point to a specific endpoint, as simply explaining alternative hypotheses not supported by the data is an insufficient approach.
  7. The conclusion that exercise will improve muscle health, while intuitive, is not supported by these data. Perhaps the authors can quantify more about the muscle properties (fiber size, fiber type) or differentiate the ability of the animal to move (differences in speed in the exercise regimen, for example).

Minor Concerns

  1. The introduction seems a bit disjointed and some smoothing of the transitions between paragraphs would make it easier to read.
  2. There are numerous incorrect words/colloquialisms throughout the manuscript. Closer proofreading for scientific writing is recommended.
  3. Text legends should be larger in all figures to make them more legible (e.g., Fig 1 B,C,E-G).
  4. The authors should present data on GDF-8 and SMAD2/3 if they are going to reference these findings in Section 3.4.
  5. The discussion refers to Fig 3 A-B, however this reviewer believes they are in fact referring to Fig 2 (line 421).

Author Response

Detailed responses to Reviewers

We appreciate the reviewer’s overall positive comments on our manuscript. In the revised manuscript, we have attempted to address all concerns raised by the reviewer. Please find the revised manuscript and figures. The resubmission includes the revised text (blue) and itemized, point-by-point responses in this letter.

Reviewer #1

In the manuscript, titled “Prolonged endurance exercise adaptations counteract doxorubicin chemotherapy-induced myotoxicity in mice”, the authors investigate the effects of exercise on mice repeatedly dosed with doxorubicin, specifically as it relates to muscle phenotype and muscle mass. While interesting, the presented data seem to contradict each other, making interpretation of the results of this study difficult. As written, this study does not reveal mechanism, and instead only suggests at some myo-protective effects that are not fully supported by the authors’ protein expression data. Perhaps some of these nuances are missing simply from a lack of context of the results, but as written the study is difficult to interpret and the impact is missing, which significantly reduced this reviewer’s enthusiasm.

Major Concerns

  1. Section 2.1 references a Figure that is not included in the manuscript draft.

Response: We appreciate the reviewer’s comment and are sorry for the omission of the figure.

We added the figure (Fig. 1A) in the revised manuscript.

  1. In Section 3.1, “it was notable that the extent of loss of the body weight and lean mass was more significant in the EXE-DOX group compared to the SED-DOX group” is unclear. Are the authors stating that there is a significant decrease in mass in EXE-DOS vs. SED-DOX (based on the reading of the figure)?

Response: We appreciate the reviewer’s comment. We replaced the vague statement with the new statement, as you suggested. The new statement “Moreover, the EXE-DOX group lost significantly higher body weight compared to the SED-DOX group during the treatment periods” was added to the revised manuscript (please see lines 255-256 in section 3.1)

  1. Are the animals the same age/size in Figure 1D? There are significant differences in the length/size (including skeletal structure) of the animals, which does not seem to make sense.

Response: We appreciate the reviewer’s comment. Please note that the figure 1D became 1E since we added figure 1A in the revised manuscript. 

The animals’ age was the same; however, because of the DOX-induced losses of body weight, lean mass, and fat mass, the smaller body sizes (including skeletal structure) in the SED-DOX and EXE-DOX were reflected in the DEXA-scanned images, which are consistent with other data (Fig. 1B, C, E, F, and G in the revised figure).   

  1. Are the CSA measurements of the entire soleus muscle, which is implied in the text (Section 3.2)? If so, including the average CSA of the myofibers would also be of interest.

Response: We appreciate the reviewer’s inquiry. We are sorry for the unclear explanation of our data; indeed, we measured the CSA of myofibers. We replaced the previous statement “the CSA of the SOL” with “the CSA of myofibers in the SOL” in the revised manuscript to avoid confusion Please see line 293 highlighted in yellow). 

  1. Quantification of the number of “ragged” fibers (Fig 2B) is necessary to determine the protective effects of exercise. While the relative increase in CS expression may indicate mitochondrial biogenesis, quantification and/or evidence of protective effects are missing. Could the authors assess the relative number of mitochondria in each muscle?

Response: We appreciate the reviewer’s comment. We quantified the number of ragged fibers and added the corresponding graph (Fig. 2C), result statement (please see lines 291-292), and figure legend (please see lines 302-303 highlighted in yellow) in the revised manuscript.

Regarding the question about assessing the relative number of mitochondria in each muscle, currently, there is no research technology available that allows us to measure the mitochondria numbers in muscles. However, transmission electron microscopy (TEM) may be a possible option to show only a snapshot of a particular mitochondrial population in a designated muscle section, not the whole muscle section. Unfortunately, the TEM was not available for the present study. 

  1. Much of the presented data seem to be contradicting each other. The authors suggest that increased FOXO3a activation would indicate an increase in proteolysis (which supports loss of muscle mass and autophagy), however then MuRF-1 expression is also diminished (which supports less muscle loss). The authors also present evidence of autophagy in the presence of DOX-EXE, but little evidence of that is visualized histologically. The authors need to supply more discussion and/or evidence to point to a specific endpoint, as simply explaining alternative hypotheses not supported by the data is an insufficient approach.

Response: We appreciate the reviewer’s concern about our data interpretation. We agree that our result statement was confusing. Yes, increased FOXO3a activation (lower levels of phosphorylated FOXO3a) would indicate an increase in proteolysis because dephosphorylated FOXO3a can translocate to nucleus to induce MuRF-1 expression. As presented in fig. 4F and G, phosphorylated FOXO3a levels were lower in the SED-DOX group than the EXE-DOX group. This means that the SED-DOX maintained active FOXO3a. As a result, MuRF-1 levels were also increased in the SED-DOX group, suggesting more proteolysis. Contrarily, FOXO3a phosphorylation (inactivated FOXO3a) levels were higher in the EXE-DOX than in the SED-DOX. As a result, MuRF-1 levels were significantly lower in the EXE-DOX than SED-DOX, suggesting less muscle loss in the EXE-DOX group. We rewrote our result statement in the revised manuscript (please see lines 473-482).

Regarding the reviewer’s comment about presenting the histological evidence of autophagy, we are very sorry that we could not provide the data in the revised manuscript because we could not conduct additional histological experiments due to the loss of frozen tissue sections caused by a freezer failure.

We strengthened our overall discussion to offset the insufficient data set that could not clearly support our hypothesis.

  1. The conclusion that exercise will improve muscle health, while intuitive, is not supported by these data. Perhaps the authors can quantify more about the muscle properties (fiber size, fiber type) or differentiate the ability of the animal to move (differences in speed in the exercise regimen, for example).

Response: We appreciate the reviewer’s comment. Again, we are very sorry that we could not provide the data that the reviewer requested in the revised manuscript because we could not conduct additional histological experiments due to the loss of frozen tissue sections caused by a freezer failure. Also, we apologize that we could not provide animal movement data since we had not conducted animal behavioral experiments. We hope the reviewer understands our excuses.

Minor Concerns

  1. The introduction seems a bit disjointed and some smoothing of the transitions between paragraphs would make it easier to read.

Response: We appreciate the reviewer’s comment. We rewrote the introduction. Please see the overall changes in the revised manuscript.

  1. There are numerous incorrect words/colloquialisms throughout the manuscript. Closer proofreading for scientific writing is recommended.

Response: We appreciate the reviewer’s comment. The errors have been corrected in the revised manuscript.

  1. Text legends should be larger in all figures to make them more legible (e.g., Fig 1 B,C,E-G).

Response: We appreciate the reviewer’s suggestion. We made the text legends legible in the revised figures.  

  1. The authors should present data on MSTN/GDF-8 and SMAD2/3 if they are going to reference these findings in Section 3.4.

Response: We thank the reviewer for the suggestion. We added MSTN and SMAD 2/3 data in the revised manuscript (please see Fig. 4A, D, and E and results’ statement in lines 337-339).    

  1. The discussion refers to Fig 3 A-B, however this reviewer believes they are in fact referring to Fig 2 (line 421).

Response: We appreciate the reviewer’s comment. According to the reviewer's indication, we corrected the figure number in the revised manuscript (please see lines 447-448). 

Reviewer 2 Report

Line 30-31: This sentence is a bit difficult to understand. I would suggest splitting the sentence.

Line 104: “Fig. 1 depicts an overview of the experimental design”. There is no figure depicting the experimental design? Such a figure would increase the comprehensibility of the experimental design.

Figure 2a and 2b: I have difficulty with the fact that the result here is based only on one representative image. Adding a statistical evaluation would make the results here clearer.

Author Response

Detailed responses to Reviewers

We appreciate the reviewers’ overall positive comments on our manuscript. In the revised manuscript, we have attempted to address all concerns raised by the reviewer. Please find the revised manuscript and figures. The resubmission includes the revised text (blue) and itemized, point-by-point responses in this letter.

Reviewer #2

  • Line 30-31: This sentence is a bit difficult to understand. I would suggest splitting the sentence.

Response: We appreciate the reviewer’s suggestion. We split the sentence into two sentences. Please see the changes in lines 29-30 highlighted in pink.

  • Line 104: “Fig. 1 depicts an overview of the experimental design”. There is no figure depicting the experimental design? Such a figure would increase the comprehensibility of the experimental design.

Response: We appreciate the reviewer’s comment. We noticed that the figure 1 was omitted in the submitted manuscript. We inserted the figure in the revised figure (Fig. 1A).

  • Figure 2a and 2b: I have difficulty with the fact that the result here is based only on one representative image. Adding a statistical evaluation would make the results here clearer.

Response: We appreciate the reviewer’s critical comment. We quantified the number of ragged fibers and added the corresponding graph (Fig.2 C), result statement (please see lines 291-292), and figure legend (please see lines 302-303 highlighted in yellow) in the revised manuscript.

Reviewer 3 Report

The manuscript focused the study of EXE-mediated protection against doxorubicin chemotherapy-induced myotoxicity in mice. The study used different biomarkers including TOP2B upregulation, satellite cells activation and myogenesis, as well as AKT-FOXO3α-MuRF-1 catabolic signaling cascades and autophagy to evaluate the effect of EXE on ameliorating the DOX-induced myotoxicity. The reviewer recommends the publication of the manuscript after the authors address the following concerns about the study:

  • Is the myotoxicity the major adverse effect of DOX when it is administrated to cancer patients? Commonly reported side effects of doxorubicin include: severe nausea and vomiting, nausea and vomiting, and alopecia. These adverse effects seem to have nothing to do with myotoxicity. The author should further explain this.
  • Mice used in this experiment were healthy mice. However, to simulate the physiological conditions of cancer patients, the author should use tumor-bearing mice in the experiment, the author should further explain this.
  • Normally, in western blot (as shown in Figure 3A, Figure 4A, 4D, Figure 5A), housekeeping proteins such beta-actin, gapdh, or tubulin was taken as internal control. But in this study, the authors seem to take ponceau as internal control. The author should further explain this.

Author Response

Detailed responses to Reviewers

We appreciate the reviewer’s overall positive comments on our manuscript. In the revised manuscript, we have attempted to address all concerns raised by the reviewer. Please find the revised manuscript and figures. The resubmission includes the revised text (blue) and itemized, point-by-point responses in this letter.

Reviewer #3

The manuscript focused the study of EXE-mediated protection against doxorubicin chemotherapy-induced myotoxicity in mice. The study used different biomarkers including TOP2B upregulation, satellite cells activation and myogenesis, as well as AKT-FOXO3α-MuRF-1 catabolic signaling cascades and autophagy to evaluate the effect of EXE on ameliorating the DOX-induced myotoxicity. The reviewer recommends the publication of the manuscript after the authors address the following concerns about the study:

  • Is the myotoxicity the major adverse effect of DOX when it is administrated to cancer patients? Commonly reported side effects of doxorubicin include: severe nausea and vomiting, and alopecia. These adverse effects seem to have nothing to do with myotoxicity. The author should further explain this.

Response: We appreciate the reviewer’s comment. We included the common side effects of DOX treatment and mentioned that DOX-induced myotoxicity is an independent concern from those side effects (please see lines 45-47 highlighted in green)

  • Mice used in this experiment were healthy mice. However, to simulate the physiological conditions of cancer patients, the author should use tumor-bearing mice in the experiment, the author should further explain this.

Response: We appreciate the reviewer’s comment. We agree that it would be the best practice to use tumor-bearing mice to simulate the physiological conditions of cancer patients. However, we chose healthy mice instead to examine the molecular mechanisms of DOX-induced myotoxicity in normal healthy skeletal muscle. Thus, it is possible that the results from nontumor-bearing mice would not represent the actual cancer physiology occurring in cancer patients. We addressed this issue in the limitations (please see lines 530-535 highlighted in green).

  • Normally, in western blot (as shown in Figure 3A, Figure 4A, 4D, Figure 5A), housekeeping proteins such beta-actin, gapdh, or tubulin was taken as internal control. But in this study, the authors seem to take ponceau as internal control. The author should further explain this.

Response: We appreciate the reviewer’s comment. We justified using Ponceau S staining over other housekeeping proteins as the loading control (please see the justification in lines 227-231 highlighted in green).
